Amorphous silica nanoparticles induce tumorigenesis via regulating ATP5H/SOD1-related oxidative stress, oxidative phosphorylation and EIF4G2/PABPC1-associated translational initiation

Xie Dongli 1
Zhou Yang 2
Luo Xiaogang 1 luoxiaogang@dhu.edu.cn
1 State Key Laboratory for Modification of Chemical Fibers and Polymer Materials, College of Materials Science and Engineering, Donghua University , Shanghai , China
2 School of Textile Science and Engineering/National Engineering Laboratory for Advanced Yarn and Clean Production, Wuhan Textile University , Wuhan , China
Anderson Todd
Electronic publication date: 2019 Mar 4
Publication date: 2019
Volume: 7
Electronic Location ID: e6455
Received 2018 Nov 19; Accepted 2019 Jan 16
Copyright: © 2019 Xie et al.
Copyright year: 2019
Copyright holder: Xie et al.
License: This is an open access article distributed under the terms of the Creative Commons Attribution License, which permits unrestricted use, distribution, reproduction and adaptation in any medium and for any purpose provided that it is properly attributed. For attribution, the original author(s), title, publication source (PeerJ) and either DOI or URL of the article must be cited.
License URL: https://creativecommons.org/licenses/by/4.0/

Keywords: Amorphous silica nanoparticles, Tumorigenesis, Oxidative stress, Translational initiation, MicroRNAs, Oxidative phosphorylation

Funding: China Postdoctoral Science Foundation 2017M621322; 2018T110324 This work was supported by a project funded by the China Postdoctoral Science Foundation (2017M621322; 2018T110324). The funders had no role in study design, data collection and analysis, decision to publish, or preparation of the manuscript.

==============================
Background

Recent studies indicate amorphous silica nanoparticles (SiNPs), one of the widely applied nanomaterials, have potential toxicity in humans and induces cell malignant transformation. However, its carcinogenic mechanisms remain poorly understood. This study’s purpose was to investigate the underlying toxic mechanisms of amorphous SiNPs on human lung epithelial cells model by using microarray data.

Methods

Microarray dataset GSE82062 was collected from Gene Expression Omnibus database, including three repeats of Beas-2B exposed to amorphous SiNPs for 40 passages and three repeats of passage-matched control Beas-2B cells. Differentially expressed genes (DEGs) were identified using linear models for microarray data method. Protein–protein interaction (PPI) network was constructed using data from the STRING database followed by module analysis. The miRwalk2 database was used to predict the underlying target genes of differentially miRNAs. Function enrichment analysis was performed using the Database for Annotation, Visualization and Integrated Discovery (DAVID) online tool.

Results

A total of 323 genes were identified as DEGs, including 280 downregulated (containing 12 pre-miRNAs) and 43 upregulated genes (containing 29 pre-miRNAs). Function enrichment indicated these genes were involved in translational initiation (i.e., eukaryotic translation initiation factor 4 gamma 2 (EIF4G2), poly (A) binding protein cytoplasmic 1 (PABPC1)), response to reactive oxygen species (i.e., superoxide dismutase 1 (SOD1)) and oxidative phosphorylation (i.e., ATP5H). PABPC1 (degree = 15), ATP5H (degree = 11) and SOD1 (degree = 8)] were proved to be hub genes after PPI-module analyses. ATP5H/SOD1 and EIF4G2/PABPC1 were overlapped with the target genes of differentially expressed pre-miR-3648/572/661 and pre-miR-4521.

Conclusions

Amorphous SiNPs may induce tumorigenesis via influencing ATP5H/SOD1-related oxidative stress, oxidative phosphorylation and EIF4G2/PABPC1-associated translational initiation which may be regulated by miR-3648/572/661 and miR-4521, respectively.

Introduction

Nanomaterials refer to those with sizes ranging from 1 to 100 nm in at least one dimension. Nanosized particles possess a number of superior physicochemical properties compared with the same material fabricated in a conventional manner, including high thermal and chemical stability, hydrophobicity, heat and electrical insulation, resistance to oxidation, good biocompatibility and minimal immunogenicity, which make them as attractive and promising candidates for a wide range of advanced applications (Liang et al., 2008; Peng et al., 2014). Silica is the most frequently used material to create nanoparticles due to its higher abundance on earth and thus a relatively low-cost for preparation. It is estimated that approximately one million tons of synthetic amorphous silica nanoparticles (SiNPs) may be produced per year worldwide to act as additives to cosmetics, drug tablets, paints, varnishes, food or deliveries for gene, protein and drugs (Fruijtier-Pölloth, 2012). The widespread application of amorphous SiNPs leads to frequent human exposure and therefore its safety is of high concern. Although amorphous SiNPs are supposedly non-carcinogenic for humans according to the classification of the International Agency for Research of Cancer (Group 3), recently published studies proposed long-term exposure of amorphous SiNPs may induce cell malignant transformation and tumorigenesis (Fontana et al., 2017; Guo et al., 2017). Therefore, how to early diagnose and prevent the development of amorphous SiNPs-induced cancer may be an underlying challenge that needs to be solved. This resulted in the requirements for understanding the molecular mechanisms of the tumor-promoting effects.

In a recent study, Guo et al. (2017) used a microarray analysis to investigate the genes significantly changed by amorphous SiNPs in human lung epithelial cells and found amorphous SiNPs may trigger tumorigenesis by influencing 821 significant differentially expressed genes (DEGs) (five upregulated and 816 downregulated) to regulate oxidative stress, oxidative phosphorylation, DNA damage, p53 and MAPK signaling pathways. However, the related mechanism leading to the development of cancer by amorphous SiNPs still remains unclear. In present study, we aim to re-analyze the microarray data established by Guo et al. (2017) via different bioinformatic approaches: the DEGs were identified by the linear models for microarray data (LIMMA) method, but not random variance model; Compared with study of Guo et al. (2017), a strict threshold was used for screening DEGs (|logFC(fold change)| > 2 & P < 0.05 vs FC > 2 & P < 0.05), which may be beneficial to obtain more crucial and verifiable genes associated with amorphous SiNPs; the whole protein–protein interaction (PPI) network for all DEGs were established, but not signal-net analysis network PPI; In addition, the miRNA-target genes interaction network was also analyzed to explore the regulatory mechanisms of DEGs and then screen key upstream targets for amorphous SiNPs, which had not previously been performed.

Materials and Methods

Microarray data

The microarray data were extracted from the gene expression omnibus (GEO) database (http://www.ncbi.nlm.nih.gov/geo/) under accession number GSE82062 (Guo et al., 2017), in which three repeats of human lung epithelial cells, Beas-2B continuously exposed to five μg/mL amorphous SiNPs for 40 passages (BeasSiNPs-P40 group; Supplemental Information 1.1, 1.2, 1.3) and three repeats of passage-matched control Beas-2B cells (Beas-P40 group) (Supplemental Information 1.4, 1.5, 1.6) were available for the analysis.

Data normalization and DEGs identification

The raw data (CEL files) of GSE82062 were downloaded from the Affymetrix Human Transcriptome Array 2.0 platform GPL17586. The raw data were preprocessed, background-corrected and summarized using robust multichip average algorithm (Irizarry et al., 2003) in the “affy” package of Bioconductor R (v3.4.1; http://www.bioconductor.org/packages/release/bioc/html/affy.html). The DEGs between BeasSiNPs-P40 and Beas-P40 groups were identified using the LIMMA method (Ritchie et al., 2015) in the Bioconductor R package (v3.4.1; http://www.bioconductor.org/packages/release/bioc/html/limma.html). The t-test was used to identify the P-value and FC was calculated. Genes were considered to be differentially expressed with the threshold value setting to |logFC| > 2 and P < 0.05.

To determine the specificity of DEGs between BeasSiNPs-P40 and Beas-P40 groups, bidirectional hierarchical clustering analysis with Euclidean distance (Szekely & Rizzo, 2005) was performed for the top 50 DEGs using the pheatmap package in R (version 1.0.8; Kolde, 2015).

PPI network construction and module analysis

The DEGs were imported into STRING database (v10.0; Search Tool for the Retrieval of Interacting Genes; https://string-db.org/) (Szklarczyk et al., 2015) to obtain the PPI data. The PPI network was constructed and visualized using Cytoscape software (v2.8; www.cytoscape.org/) (Kohl, Wiese & Warscheid, 2011). The hub genes with more interactions with other partners (degree) were selected and plotted with ggplot2 in R package (v3.4.1; R Core Team, 2017).

Furthermore, the Molecular Complex Detection (MCODE) plugin of Cytoscape software was also employed to identify functionally related and highly interconnected clusters from the PPI network with degree cutoff of 6, node score cutoff of 0.2, k-core of 5 and maximum depth of 100 (ftp://ftp.mshri.on.ca/pub/BIND/Tools/MCODE) (Bader & Hogue, 2003). Sub-modules were defined to be significant with MCODE score ≥ 4 and nodes ≥ 6.

miRNA-target genes interaction prediction

Because there were some pre-miRNAs included in the DEGs, the pre-miRNA–mRNA interaction was also investigated to explore the potential regulatory mechanisms of mRNAs. mRNA targets of differentially expressed miRNAs (DE-miRNAs) were predicted using the miRWalk database (v2.0; http://zmf.umm.uni-heidelberg.de/apps/zmf/mirwalk2/) (Dweep & Gretz, 2015), which were then overlapped with the DEGs to obtain DE-miRNA–DEGs interaction relationships. Then, the DE-miRNA–DEG interaction network was constructed and visualized using Cytoscape software (Kohl, Wiese & Warscheid, 2011).

Function enrichment analysis

Gene ontology (GO) biological process terms, Kyoto Encyclopedia of Genes and Genomes (KEGG), and BioCarta pathways were also enriched for all DEGs, DEGs in PPI and miRNA–DEG interaction networks by the Database for Annotation, Visualization and Integrated Discovery (DAVID) online tool (v6.8; http://david.abcc.ncifcrf.gov) (Dennis et al., 2003). P-value < 0.05 was chosen as the threshold value for functional enrichment analyses.

Results

Identification of DEGs

After preprocessing and data normalization (gene expression in all samples are shown in Supplemental Information 2), DEGs between BeasSiNPs-P40 and Beas-P40 groups were identified by the LIMMA method. As a result, 323 genes were considered to be differentially expressed due to exceeding the difference threshold (|logFC| > 2 and P < 0.05), including 280 downregulated (containing 12 pre-miRNAs) and 43 upregulated genes (containing 29 pre-miRNAs) (Fig. 1), which was fewer than the study results of Guo et al. (2017). All the DEGs are listed in in Supplemental Information 3.

Figure 1 Heat map of differentially expressed genes identified between human lung epithelial cell Beas-2B exposed to amorphous silica nanoparticles or not.

High level of expression was indicated as red and low level was in green. Heatmap was established based on normalized expression values of significantly changed mRNAs. The expression values are depicted in line with the color scale. The intensity increases from green to red. Each column represents one sample, and each row indicates a transcript.

Function enrichment analysis of DEGs

The above differential genes were subjected to the online tool DAVID for function enrichment analysis. As a result, 49 GO biological process terms were enriched for downregulated DEGs, including response to reactive oxygen species (ROS) (i.e., superoxide dismutase 1 (SOD1)). Only two GO biological process terms were enriched for upregulated DEGs, mainly involving in protein translation (i.e., eukaryotic translation initiation factor 4 gamma 2 (EIF4G2); poly (A) binding protein cytoplasmic 1 (PABPC1)) (Table 1).

Table 1 Gene ontology (GO) biological process terms analysis for all differentially expressed genes.

	Term	P-value	Genes	
Down	GO:0006334∼nucleosome assembly	2.32E-16	HIST1H2BB, HIST1H3J, HIST1H4L, HIST1H1E, HIST1H1C, HIST1H2BF, HIST1H2BO, HIST1H2BK, HIST1H4B, HIST1H2BI, HIST1H3A, HIST1H4E, HIST1H3B, HIST1H4C, HIST1H3D, HIST1H4D, HIST1H3F	
	GO:0051290∼protein heterotetramerization	5.54E-15	HIST1H3J, HIST1H4L, HIST1H4B, XRCC6, HIST1H3A, HIST1H3B, HIST1H4E, HIST1H3D, HIST1H4C, HIST1H4D, HIST1H3F, ANXA2	
GO:0032200∼telomere organization	1.51E-13	HIST1H3J, HIST1H4L, HIST1H4B, HIST1H3A, HIST1H3B, HIST1H4E, HIST1H3D, HIST1H4C, HIST1H4D, HIST1H3F	
GO:0045815∼positive regulation of gene expression, epigenetic	5.82E-13	HIST1H3J, HIST1H4L, POLR2L, POLR2K, HIST1H4B, HIST1H3A, HIST1H3B, HIST1H4E, HIST1H3D, HIST1H4C, HIST1H4D, HIST1H3F	
GO:0006335∼DNA replication-dependent nucleosome assembly	8.76E-13	HIST1H3J, HIST1H4L, HIST1H4B, HIST1H3A, HIST1H3B, HIST1H4E, HIST1H3D, HIST1H4C, HIST1H4D, HIST1H3F	
GO:0000183∼chromatin silencing at rDNA	3.77E-12	HIST1H3J, HIST1H4L, HIST1H4B, HIST1H3A, HIST1H3B, HIST1H4E, HIST1H3D, HIST1H4C, HIST1H4D, HIST1H3F	
GO:0045814∼negative regulation of gene expression, epigenetic	7.03E-11	HIST1H3J, HIST1H4L, HIST1H4B, HIST1H3A, HIST1H3B, HIST1H4E, HIST1H3D, HIST1H4C, HIST1H4D, HIST1H3F	
GO:0031047∼gene silencing by RNA	4.04E-10	HIST1H3J, HIST1H4L, POLR2L, POLR2K, HIST1H4B, HIST1H3A, HIST1H3B, HIST1H4E, HIST1H3D, HIST1H4C, HIST1H4D, HIST1H3F	
GO:0044267∼cellular protein metabolic process	7.84E-10	HIST1H3J, HIST1H4L, HIST1H4B, TGFBI, HIST1H3A, HIST1H3B, HIST1H4E, HIST1H3D, HIST1H4C, HIST1H4D, HIST1H3F, B2M	
GO:0006614∼SRP-dependent cotranslational protein targeting to membrane	2.29E-08	RPS28, RPS29, RPL41, RPS15, RPL26, RPS13, RPS15A, RPS4Y1, RPL24, RPS5	
GO:0098609∼cell-cell adhesion	7.12E-08	HIST1H3J, CHMP5, EEF2, RPL24, PDLIM1, SFN, ANXA2, KRT18, HIST1H3A, HIST1H3B, HIST1H3D, CNN2, HIST1H3F, ENO1	
GO:0019083∼viral transcription	1.06E-07	RPS28, RPS29, RPL41, RPS15, RPL26, RPS13, RPS15A, RPS4Y1, RPL24, RPS5	
GO:0000184∼nuclear-transcribed mRNA catabolic process, nonsense-mediated decay	1.80E-07	RPS28, RPS29, RPL41, RPS15, RPL26, RPS13, RPS15A, RPS4Y1, RPL24, RPS5	
GO:0006413∼translational initiation	5.98E-07	RPS28, RPS29, RPL41, RPS15, RPL26, RPS13, RPS15A, RPS4Y1, RPL24, RPS5	
GO:0060968∼regulation of gene silencing	7.87E-07	HIST1H3J, HIST1H3A, HIST1H3B, HIST1H3D, HIST1H3F	
GO:0006364∼rRNA processing	3.14E-06	RPS28, RPS29, RPL41, RRP36, RPS15, RPL26, RPS13, RPS15A, RPS4Y1, RPL24, RPS5	
GO:0006303∼double-strand break repair via nonhomologous end joining	5.72E-06	HIST1H4L, HIST1H4B, XRCC6, HIST1H4E, BABAM1, HIST1H4C, HIST1H4D	
GO:0045653∼negative regulation of megakaryocyte differentiation	7.02E-06	HIST1H4L, HIST1H4B, HIST1H4E, HIST1H4C, HIST1H4D	
GO:0034080∼CENP-A containing nucleosome assembly	1.33E-05	HIST1H4L, HIST1H4B, HIST1H4E, CENPW, HIST1H4C, HIST1H4D	
GO:0006412∼translation	1.37E-05	RPS28, RPS29, RPL41, RPS15, RPL26, RPS13, RPS15A, RPS4Y1, RPL24, MRPS7, RPS5	
GO:0006342∼chromatin silencing	1.67E-05	HIST1H2AB, HIST2H2AB, HIST2H2AA4, HIST2H2AC, HIST1H2AE, HIST1H2AM	
GO:0016233∼telomere capping	1.98E-05	HIST1H4L, HIST1H4B, HIST1H4E, HIST1H4C, HIST1H4D	
GO:0006336∼DNA replication-independent nucleosome assembly	3.28E-05	HIST1H4L, HIST1H4B, HIST1H4E, HIST1H4C, HIST1H4D	
GO:0006352∼DNA-templated transcription, initiation	1.22E-04	HIST1H4L, HIST1H4B, HIST1H4E, HIST1H4C, HIST1H4D	
GO:1904837∼beta-catenin-TCF complex assembly	2.47E-04	HIST1H4L, HIST1H4B, HIST1H4E, HIST1H4C, HIST1H4D	
GO:0000028∼ribosomal small subunit assembly	3.17E-04	RPS28, RPS15, MRPS7, RPS5	
GO:0006123∼mitochondrial electron transport, cytochrome c to oxygen	3.71E-04	COX7B, COX8A, COX6B1, COX6A1	
GO:1902600∼hydrogen ion transmembrane transport	9.45E-04	UQCR10, COX7B, COX8A, COX6B1, COX6A1	
GO:0043154∼negative regulation of cysteine-type endopeptidase activity involved in apoptotic process	1.50E-03	PRDX5, SFN, THBS1, IFI6, DHCR24	
GO:0019731∼antibacterial humoral response	3.81E-03	HIST1H2BK, HIST1H2BF, HIST1H2BI, B2M	
GO:0042274∼ribosomal small subunit biogenesis	5.69E-03	RPS28, RRP36, RPS15	
GO:0002576∼platelet degranulation	6.37E-03	PSAP, THBS1, SOD1, SRGN, FN1	
GO:0002227∼innate immune response in mucosa	1.36E-02	HIST1H2BK, HIST1H2BF, HIST1H2BI	
GO:0043086∼negative regulation of catalytic activity	1.66E-02	OAZ1, NQO1, ANXA2, ANXA2P2	
GO:0050434∼positive regulation of viral transcription	1.81E-02	POLR2L, POLR2K, SUPT4H1	
GO:0050830∼defense response to Gram-positive bacterium	2.30E-02	HIST1H2BK, HIST1H2BF, HIST1H2BI, B2M	
GO:0006368∼transcription elongation from RNA polymerase II promoter	2.37E-02	POLR2L, POLR2K, ELOF1, SUPT4H1	
GO:0033490∼cholesterol biosynthetic process via lathosterol	2.83E-03	DHCR7, DHCR24	
GO:0033489∼cholesterol biosynthetic process via desmosterol	2.83E-02	DHCR7, DHCR24	
GO:0007568∼aging	3.07E-02	CCL2, EEF2, SOD1, NQO1, APEX1	
GO:0000302∼response to reactive oxygen species	3.16E-02	TXN, PRDX5, SOD1	
GO:0001895∼retina homeostasis	3.31E-02	ACTG1, SOD1, B2M	
GO:0045471∼response to ethanol	3.94E-02	CCL2, EEF2, SOD1, NQO1	
GO:0006356∼regulation of transcription from RNA polymerase I promoter	4.21E-02	POLR2L, POLR2K	
GO:0098532∼histone H3-K27 trimethylation	4.21E-02	HIST1H1E, HIST1H1C	
GO:0007596∼blood coagulation	4.30E-02	HIST1H3J, HIST1H3A, HIST1H3B, HIST1H3D, HIST1H3F	
GO:0009615∼response to virus	4.43E-02	IFITM2, IFITM3, RPS15A, ENO1	
GO:0043066∼negative regulation of apoptotic process	4.48E-02	KRT18, PLK2, PLK1, TMBIM4, PRDX5, THBS1, NQO1, DHCR24	
GO:1900121∼negative regulation of receptor binding	4.90E-02	ANXA2, B2M	
Up	GO:0045727∼positive regulation of translation	1.57E-02	EIF4G2, PABPC1	
GO:0006413∼translational initiation	4.01E-02	EIF4G2, PABPC1	

Kyoto Encyclopedia of Genes and Genomes pathway analyses were also performed by DAVID software. In line with the GO terms, the downregulated DEGs were predicted to participate in oxidative stress related metabolism pathways, such as oxidative phosphorylation (i.e., ATP5H, ATP synthase, H+ transporting, mitochondrial F0 complex, subunit D), while upregulated DEGs were enriched for Regulation of eIF4e and p70 S6 Kinase (i.e., EIF4G2, PABPC1) (Table 2).

Table 2 KEGG pathway enrichment analysis for all differentially expressed genes.

	Term	P-value	Genes	
Down	hsa05322: Systemic lupus erythematosus	2.06E-16	HIST1H2AB, HIST1H2BB, HIST1H3J, HIST1H4L, HIST2H2AA4, HIST1H2BF, HIST1H2AE, HIST1H2BO, HIST2H2AB, HIST1H2BK, HIST1H4B, HIST2H2AC, HIST1H2BI, HIST1H3A, HIST1H4E, HIST1H3B, HIST1H3D, HIST1H4C, HIST1H4D, HIST1H3F, HIST1H2AM	
hsa05034: Alcoholism	5.44E-14	HIST1H2AB, HIST1H2BB, HIST1H3J, HIST1H4L, HIST2H2AA4, HIST1H2BF, HIST1H2AE, HIST1H2BO, HIST2H2AB, HIST1H2BK, HIST1H4B, HIST2H2AC, HIST1H2BI, HIST1H3A, HIST1H4E, HIST1H3B, HIST1H3D, HIST1H4C, HIST1H4D, HIST1H3F, HIST1H2AM	
hsa03010: Ribosome	9.39E-06	RPS28, RPS29, RPL41, RPS15, RPL26, RPS13, RPS15A, RPS4Y1, RPL24, MRPS7, RPS5	
hsa05016: Huntington’s disease	1.80E-04	UQCR10, NDUFS5, POLR2L, POLR2K, COX7B, COX8A, COX6B1, COX6A1, SOD1, NDUFA1, ATP5H	
hsa00190: Oxidative phosphorylation	3.14E-04	UQCR10, NDUFS5, ATP6V0E1, COX7B, COX8A, COX6B1, COX6A1, NDUFA1, ATP5H	
hsa05203: Viral carcinogenesis	1.32E-03	HIST1H2BO, HIST1H2BB, HIST1H4L, HIST1H2BK, HIST1H2BF, HIST1H4B, HIST1H2BI, HIST1H4E, HIST1H4C, HIST1H4D	
hsa05010: Alzheimer’s disease	1.47E-03	UQCR10, NDUFS5, COX7B, COX8A, COX6B1, COX6A1, PSENEN, NDUFA1, ATP5H	
hsa05012: Parkinson’s disease	2.41E-03	UQCR10, NDUFS5, COX7B, COX8A, COX6B1, COX6A1, NDUFA1, ATP5H	
hsa04932: Non-alcoholic fatty liver disease (NAFLD)	1.36E-02	UQCR10, NDUFS5, COX7B, COX8A, COX6B1, COX6A1, NDUFA1	
hsa04260: Cardiac muscle contraction	1.62E-02	UQCR10, COX7B, COX8A, COX6B1, COX6A1	
hsa03008: Ribosome biogenesis in eukaryotes	2.64E-02	SNORD3A, SNORD3C, SNORD3B-1, SNORD3B-2, NOP10	
Up	BIOCARTA
h_eif4Pathway: Regulation of eIF4e and p70 S6 Kinase	1.48E-02	EIF4G2, PABPC1	
Note:

KEGG, Kyoto encyclopedia of genes and genomes.

PPI network construction

After mapping the DEGs into the PPI data, a DEG-related PPI network was constructed (Fig. 2A), including 92 nodes (86 downregulated and six upregulated) and 291 edges (interaction relationships, such as EIF4G2-PABPC1, SOD1-ATP5H) (Supplemental Information 4). After calculating the degree, PABPC1 (degree = 15), ATP5H (degree = 11) and SOD1 (degree = 8) were suggested to be hub genes (Fig. 2B).

Figure 2 Protein–protein interaction network to screen crucial genes.

(A) Protein–protein interaction network of differentially expressed genes between human lung epithelial cell Beas-2B exposed to amorphous silica nanoparticles or not. Downregulated genes were indicated as orange and upregulated genes were in green. (B) Proteins rank according their interaction pairs in the protein–protein interaction network.

Following cluster analysis using the MCODE algorithm, three significant modules were obtained (Fig. 3). Function enrichment analysis showed that ATP5H was included in module 3 and participated in Oxidative phosphorylation pathway (Table 3).

Figure 3 Modules obtained from PPI network.

Orange, downregulated genes; green, upregulated genes. (A) Module 1; (B) module 2; (C) module 3.

Table 3 KEGG pathway enrichment analysis for DEGs in modules.

Module	Term	P-value	Genes	
1	hsa03010: Ribosome	8.25E-17	RPS28, RPL41, RPS29, RPS15, RPL26, RPS4Y1, RPS13, RPS15A, RPL24, RPS5	
2	hsa05322: Systemic lupus erythematosus	9.23E-09	HIST1H2BO, HIST1H2BB, HIST1H2BK, HIST2H2AC, HIST1H3B, HIST1H2AM, HIST3H3	
3	hsa05012: Parkinson’s disease	5.44E-12	NDUFS5, UQCR10, COX8A, COX7B, COX6B1, COX6A1, NDUFA1, ATP5H	
	hsa00190: Oxidative phosphorylation	6.08E-12	NDUFS5, UQCR10, COX8A, COX7B, COX6B1, COX6A1, NDUFA1, ATP5H	
	hsa05010: Alzheimer’s disease	3.06E-11	NDUFS5, UQCR10, COX8A, COX7B, COX6B1, COX6A1, NDUFA1, ATP5H	
	hsa05016: Huntington’s disease	6.21E-11	NDUFS5, UQCR10, COX8A, COX7B, COX6B1, COX6A1, NDUFA1, ATP5H	
	hsa04260: Cardiac muscle contraction	1.73E-06	UQCR10, COX8A, COX7B, COX6B1, COX6A1	
Note:

DEGs, differentially expressed genes; KEGG, Kyoto encyclopedia of genes and genomes.

miRNA–mRNA interaction network

A total of 18,896 target genes were predicted for the 41 DE-miRNAs by using the miRWalk database, 130 of which (including 124 downregulated and six upregulated) were found to be shared with the DEGs. Then, a miRNA–mRNA interaction network was constructed by using the 130 target genes and their corresponding 24 miRNAs, resulting in 1,615 and 17 interrelationship pairs for the 18 upregulated miRNAs and six downregulated miRNAs, respectively (Fig. 4).

Figure 4 A miRNA-gene interaction network.

Blue diamond, upregulated miRNAs; yellow diamond, downregulated miRNAs; red oval, downregulated differentially expressed genes; green, upregulated differentially expressed genes.

Function enrichment analysis also demonstrated these 130 target genes were involved in GO terms of response to reactive oxygen species (hsa-miR-3155b/3648/4522/4523/4533/4634/4734/572/661/933/1470-SOD1), and translational initiation (miR-5095/548ai/622/4521-EIF4G2, miR-4521-PABPC1) (Table 4) and KEGG pathways of Oxidative phosphorylation (hsa-miR-1976/3155/3648/4461/4522/4523/4533/4634/4734/572/661/933-ATP5H) (Table 5).

Table 4 Gene ontology (GO) biological process terms analysis for DEGs in miRNA–mRNA regulatory network.

Term	P-value	Genes	
GO: 0006334∼nucleosome assembly	1.02E-17	HIST1H2BB, HIST1H3J, HIST1H4L, HIST1H1E, HIST1H1C, HIST1H2BF, HIST1H2BO, HIST1H2BK, HIST1H4B, HIST1H2BI, HIST1H3A, HIST1H4E, HIST1H3B, HIST1H3D, HIST1H4C, HIST1H4D, HIST1H3F, HIST3H3	
GO: 0051290∼protein heterotetramerization	1.02E-16	HIST1H3J, HIST1H4L, XRCC6, ANXA2, HIST1H4B, HIST1H3A, HIST1H4E, HIST1H3B, HIST1H4C, HIST1H3D, HIST1H4D, HIST3H3, HIST1H3F	
GO: 0032200∼telomere organization	1.63E-13	HIST1H3J, HIST1H4L, HIST1H4B, HIST1H3A, HIST1H3B, HIST1H4E, HIST1H3D, HIST1H4C, HIST1H4D, HIST1H3F	
GO: 0045815∼positive regulation of gene expression, epigenetic	6.38E-13	HIST1H3J, HIST1H4L, POLR2L, POLR2K, HIST1H4B, HIST1H3A, HIST1H3B, HIST1H4E, HIST1H3D, HIST1H4C, HIST1H4D, HIST1H3F	
GO: 0006335∼DNA replication-dependent nucleosome assembly	9.45E-13	HIST1H3J, HIST1H4L, HIST1H4B, HIST1H3A, HIST1H3B, HIST1H4E, HIST1H3D, HIST1H4C, HIST1H4D, HIST1H3F	
GO: 0000183∼chromatin silencing at rDNA	4.07E-12	HIST1H3J, HIST1H4L, HIST1H4B, HIST1H3A, HIST1H3B, HIST1H4E, HIST1H3D, HIST1H4C, HIST1H4D, HIST1H3F	
GO: 0031047∼gene silencing by RNA	2.43E-11	HIST1H3J, HIST1H4L, POLR2L, POLR2K, HIST1H4B, HIST1H3A, HIST1H4E, HIST1H3B, HIST1H4C, HIST1H3D, HIST1H4D, PABPC1, HIST1H3F	
GO: 0045814∼negative regulation of gene expression, epigenetic	7.57E-11	HIST1H3J, HIST1H4L, HIST1H4B, HIST1H3A, HIST1H3B, HIST1H4E, HIST1H3D, HIST1H4C, HIST1H4D, HIST1H3F	
GO: 0044267∼cellular protein metabolic process	8.57E-10	HIST1H3J, HIST1H4L, HIST1H4B, TGFBI, HIST1H3A, HIST1H3B, HIST1H4E, HIST1H3D, HIST1H4C, HIST1H4D, HIST1H3F, B2M	
GO: 0006413∼translational initiation	4.23E-09	EIF4G2, RPS28, RPS29, RPL41, RPS15, RPL26, RPS13, RPS15A, RPS4Y1, RPL24, PABPC1, RPS5	
GO: 0098609∼cell-cell adhesion	9.43E-09	HIST1H3J, CHMP5, PDLIM1, EEF2, RPL24, SFN, ANXA2, EIF4G2, KRT18, HIST1H3A, HIST1H3B, HIST1H3D, CNN2, HIST1H3F, ENO1	
GO: 0000184∼nuclear-transcribed mRNA catabolic process, nonsense-mediated decay	1.42E-08	RPS28, RPS29, RPL41, RPS15, RPL26, RPS13, RPS15A, RPS4Y1, RPL24, PABPC1, RPS5	
GO: 0006614∼SRP-dependent cotranslational protein targeting to membrane	2.47E-08	RPS28, RPS29, RPL41, RPS15, RPL26, RPS13, RPS15A, RPS4Y1, RPL24, RPS5	
GO: 0019083∼viral transcription	1.14E-07	RPS28, RPS29, RPL41, RPS15, RPL26, RPS13, RPS15A, RPS4Y1, RPL24, RPS5	
GO: 0006303∼double-strand break repair via nonhomologous end joining	3.36E-07	HIST1H4L, HIST1H4B, XRCC6, HIST1H4E, BABAM1, HIST1H4C, HIST1H4D, HIST3H3	
GO: 0016233∼telomere capping	5.42E-07	HIST1H4L, HIST1H4B, HIST1H4E, HIST1H4C, HIST1H4D, HIST3H3	
GO: 0060968∼regulation of gene silencing	8.14E-07	HIST1H3J, HIST1H3A, HIST1H3B, HIST1H3D, HIST1H3F	
GO: 0006364∼rRNA processing	3.38E-06	RPS28, RPS29, RPL41, RRP36, RPS15, RPL26, RPS13, RPS15A, RPS4Y1, RPL24, RPS5	
GO: 0045653∼negative regulation of megakaryocyte differentiation	7.26E-06	HIST1H4L, HIST1H4B, HIST1H4E, HIST1H4C, HIST1H4D	
GO: 0034080∼CENP-A containing nucleosome assembly	1.38E-05	HIST1H4L, HIST1H4B, HIST1H4E, CENPW, HIST1H4C, HIST1H4D	
GO: 0006412∼translation	1.48E-05	RPS28, RPS29, RPL41, RPS15, RPL26, RPS13, RPS15A, RPS4Y1, RPL24, MRPS7, RPS5	
GO: 0006342∼chromatin silencing	1.73E-05	HIST1H2AB, HIST2H2AB, HIST2H2AA4, HIST2H2AC, HIST1H2AE, HIST1H2AM	
GO: 0006336∼DNA replication-independent nucleosome assembly	3.39E-05	HIST1H4L, HIST1H4B, HIST1H4E, HIST1H4C, HIST1H4D	
GO: 0006352∼DNA-templated transcription, initiation	1.26E-04	HIST1H4L, HIST1H4B, HIST1H4E, HIST1H4C, HIST1H4D	
GO: 1904837∼beta-catenin-TCF complex assembly	2.55E-04	HIST1H4L, HIST1H4B, HIST1H4E, HIST1H4C, HIST1H4D	
GO: 0000028∼ribosomal small subunit assembly	3.25E-04	RPS28, RPS15, MRPS7, RPS5	
GO: 0006123∼mitochondrial electron transport, cytochrome c to oxygen	3.80E-04	COX7B, COX8A, COX6B1, COX6A1	
GO: 0045727∼positive regulation of translation	5.72E-04	EIF4G2, EEF2, QARS, PABPC1, THBS1	
GO: 1902600∼hydrogen ion transmembrane transport	9.75E-04	UQCR10, COX7B, COX8A, COX6B1, COX6A1	
GO: 0043154∼negative regulation of cysteine-type endopeptidase activity involved in apoptotic process	1.55E-03	PRDX5, SFN, THBS1, IFI6, DHCR24	
GO: 0019731∼antibacterial humoral response	3.90E-03	HIST1H2BK, HIST1H2BF, HIST1H2BI, B2M	
GO: 0042274∼ribosomal small subunit biogenesis	5.79E-03	RPS28, RRP36, RPS15	
GO: 0002227∼innate immune response in mucosa	1.39E-02	HIST1H2BK, HIST1H2BF, HIST1H2BI	
GO: 0050434∼positive regulation of viral transcription	1.84E-02	POLR2L, POLR2K, SUPT4H1	
GO: 0050830∼defense response to Gram-positive bacterium	2.35E-02	HIST1H2BK, HIST1H2BF, HIST1H2BI, B2M	
GO: 0006368∼transcription elongation from RNA polymerase II promoter	2.42E-02	POLR2L, POLR2K, ELOF1, SUPT4H1	
GO: 0033489∼cholesterol biosynthetic process via desmosterol	2.85E-02	DHCR7, DHCR24	
GO: 0033490∼cholesterol biosynthetic process via lathosterol	2.85E-02	DHCR7, DHCR24	
GO: 0007568∼aging	3.15E-02	CCL2, EEF2, SOD1, NQO1, APEX1	
GO: 0000302∼response to reactive oxygen species	3.21E-02	TXN, PRDX5, SOD1	
GO: 0001895∼retina homeostasis	3.36E-02	ACTG1, SOD1, B2M	
GO: 0002576∼platelet degranulation	3.84E-02	PSAP, THBS1, SOD1, SRGN	
GO: 0043488∼regulation of mRNA stability	3.84E-02	PSMB3, HSPA1B, PABPC1, APEX1	
GO: 0045471∼response to ethanol	4.03E-02	CCL2, EEF2, SOD1, NQO1	
GO: 0006356∼regulation of transcription from RNA polymerase I promoter	4.25E-02	POLR2L, POLR2K	
GO: 0098532∼histone H3-K27 trimethylation	4.25E-02	HIST1H1E, HIST1H1C	
GO: 0007596∼blood coagulation	4.41E-02	HIST1H3J, HIST1H3A, HIST1H3B, HIST1H3D, HIST1H3F	
GO: 0009615∼response to virus	4.52E-02	IFITM2, IFITM3, RPS15A, ENO1	
GO: 0043066∼negative regulation of apoptotic process	4.65E-02	KRT18, PLK2, PLK1, TMBIM4, PRDX5, THBS1, NQO1, DHCR24	
GO: 1900121∼negative regulation of receptor binding	4.94E-02	ANXA2, B2M	

Table 5 KEGG pathway enrichment analysis for DEGs in miRNA–mRNA regulatory network.

Term	P-value	Genes	
hsa05322: Systemic lupus erythematosus	5.48E-18	HIST1H2AB, HIST1H2BB, HIST1H3J, HIST1H4L, HIST2H2AA4, HIST1H2BF, HIST1H2AE, HIST1H2BO, HIST2H2AB, HIST1H2BK, HIST1H4B, HIST2H2AC, HIST1H2BI, HIST1H3A, HIST1H4E, HIST1H3B, HIST1H3D, HIST1H4C, HIST1H4D, HIST1H3F, HIST1H2AM, HIST3H3	
hsa05034: Alcoholism	2.00E-15	HIST1H2AB, HIST1H2BB, HIST1H3J, HIST1H4L, HIST2H2AA4, HIST1H2BF, HIST1H2AE, HIST1H2BO, HIST2H2AB, HIST1H2BK, HIST1H4B, HIST2H2AC, HIST1H2BI, HIST1H3A, HIST1H4E, HIST1H3B, HIST1H3D, HIST1H4C, HIST1H4D, HIST1H3F, HIST1H2AM, HIST3H3	
hsa03010: Ribosome	6.90E-06	RPS28, RPS29, RPL41, RPS15, RPL26, RPS13, RPS15A, RPS4Y1, RPL24, MRPS7, RPS5	
hsa05016: Huntington’s disease	6.46E-04	UQCR10, NDUFS5, POLR2L, POLR2K, COX7B, COX8A, COX6B1, COX6A1, SOD1, ATP5H	
hsa05203: Viral carcinogenesis	1.03E-03	HIST1H2BO, HIST1H2BB, HIST1H4L, HIST1H2BK, HIST1H2BF, HIST1H4B, HIST1H2BI, HIST1H4E, HIST1H4C, HIST1H4D	
hsa00190: Oxidative phosphorylation	1.36E-03	UQCR10, NDUFS5, ATP6V0E1, COX7B, COX8A, COX6B1, COX6A1, ATP5H	
hsa05010: Alzheimer’s disease	5.06E-03	UQCR10, NDUFS5, COX7B, COX8A, COX6B1, COX6A1, PSENEN, ATP5H	
hsa05012: Parkinson’s disease	8.75E-03	UQCR10, NDUFS5, COX7B, COX8A, COX6B1, COX6A1, ATP5H	
hsa04260: Cardiac muscle contraction	1.45E-02	UQCR10, COX7B, COX8A, COX6B1, COX6A1	
hsa04932: Non-alcoholic fatty liver disease (NAFLD)	4.12E-02	UQCR10, NDUFS5, COX7B, COX8A, COX6B1, COX6A1	

Discussion

After PPI hub genes and module analysis as well as miRNA target gene prediction, our present study preliminarily suggests downregulated ATP5H, SOD1 and upregulated EIF4G2, PABPC1 may be especially important genes involved in amorphous SiNPs-mediated tumor initiation. EIF4G2 and PABPC1 may be involved in the progression of cancer by affecting translational initiation, while SOD1 and ATP5H may participate in carcinogenesis via influencing oxidative stress and oxidative phosphorylation. Although genes were not similar, these enriched pathways seemed to be in line with the study of Guo et al. (2017), further indicating these biological processes may be crucial.

Recently, accumulating evidence has indicated exposure to amorphous SiNPs induces oxidative stress in cells or organs (Guo et al., 2015; Wu et al., 2016; Nemmar et al., 2016). For example, Jiang et al. (2016) incubated the erythrocytes with SiNPs and found the oxidative damage biomarker malondialdehyde was significantly increased, while the activity of antioxidant superoxide dismutase (SOD) was decreased. Di Cristo et al. (2016) showed exposure of SiNPs to macrophages elicited a greater oxidative stress than was assessed from heme oxygenase-1 induction and ROS production. Nemmar et al. (2016) observed intraperitoneal administration of SiNPs induced significantly increased lipid peroxidation in the lung, liver, kidney and brain of mice, displaying reduced SOD and catalase activities. Consistent with these studies, our study also found anti-oxidative gene SOD1 was significantly downregulated after SiNPs treatment. The increased oxidative stress was reported to induce the switch of glucose metabolism from oxidative phosphorylation to aerobic glycolysis (the Warburg Effect) to promote excessive proliferation and growth of cells, leading to the development and progression of cancer (Pani, Galeotti & Chiarugi, 2010; Molavian, Kohandel & Sivaloganathan, 2016). ATP synthase is an enzyme to be responsible for the synthesis of abundant ATP in oxidative phosphorylation process. The expression of ATP synthase may be reduced due to the decreased oxidative phosphorylation. As expected, Feichtinger et al. (2018) found significantly reduced levels of ATP Synthase Subunit ATP5F1A was correlated with earlier-onset prostate cancer. Shin et al. (2005) used the two-dimensional gel electrophoresis data to demonstrate the expression and activity of ATP synthase were lower in 5-FU-resistant cells compared with parent cancer cells. Inhibition of ATP synthase by oligomycin A or siRNA transfection strongly antagonized 5-FU-induced suppression of cell proliferation and increased cell viability. Similarly, the study of Song et al. (2018) indicated loss of ATP5H conferred a stem-like, invasive phenotype to tumor cells as well as multimodal resistance to immunotherapy, chemotherapy and radiotherapy. Also, reduced levels of ATP synthase were associated with poor prognosis in cancer patients (Song et al., 2018). In agreement with these studies, we also found ATP5H could interact with SOD1 and was downregulated in malignant transformation after SiNPs treatment.

In accordance with our hypothesis that the increased oxidative stress and aerobic glycolysis may accelerate the cell proliferation, we also found translational initiation was abnormally activated in human lung epithelial cells after SiNPs exposure, which may, on one hand, promote cell division and on the other hand, maintains cell survival (Pyronnet & Sonenberg, 2001). Translation process requires the protein complex known as eukaryotic initiation factor 4F (eIF4F) which consists of cap-binding protein eIF4E, scaffolding protein eIF4G and ATP-dependent RNA helicase eIF4A (Gingras, Raught & Sonenberg, 1999). Hereby, the expression of translation protein related genes may be upregulated in malignant transformation, which have been confirmed in several cancers because downregulation of eIF4GII was reported to decrease cell proliferation, but induces cellular senescence (Emmrich et al., 2016; Xie et al., 2017). During translation process in eukaryotic cells, nonsense mutations (premature stop codon) may be present, which disrupts production of full-length, functional proteins (such as tumor suppressor gene p53) and thus may induce the development of various diseases, including cancers (Kashofer et al., 2017). Nonsense-mediated mRNA decay (NMD) represents a surveillance mechanism that eliminates transcripts with nonsense mutations and prevents cancer development. In contrast, inhibition of NMD may result in the initiation of cancer (Cao et al., 2017). Recent studies showed PABPC1 can interact with eIF4G to inhibit NMD (Fatscher et al., 2014; Peixeiro et al., 2012). Thus, the upregulation of PABPC1 may be one underlying reason for tumor formation. This hypothesis has been demonstrated in the gastric carcinoma (Zhu et al., 2015) and hepatocellular carcinoma (Zhang et al., 2015b) samples. As expected, our present study also demonstrated SiNPs may induce the tumorigenesis of Beas-2B cells by upregulating EIF4G2 and PABPC1. These two genes could interact with each other.

More interestingly, our study showed the downregulation of ATP5H and SOD1 may be resulted from the upregulation of their common upstream miR-3648/572/661, while the downregulated miR-4521 may lead to the upregulation of the transcription of EIF4G2 and PABPC1. Although all these were the potential mechanisms firstly obtained for the carcinogenicity of amorphous SiNPs, recent studies on the miR-3648/572/661 and miR-4521 may indirectly demonstrate their importance in cancer development. For example, Rashid et al. (2017) demonstrated overexpression of miR-3648 promoted the growth of HeLa cells, while opposite results were obtained when miR-3648 was inhibited by antagomir. The mechanism of miR-3648 for cancer progression was to suppress the expression of a tumor suppressor gene adenomatous polyposis coli 2. miR-572 was also found to be highly expressed in human ovarian cancer tissues and cell lines. Ectopic overexpression of miR-572 promoted ovarian cancer cell proliferation and cell cycle progression in vitro and tumorigenicity in vivo by inhibiting its direct target suppressor of cytokine signaling 1 and cyclin-dependent kinase inhibitor 1A (p21KIP). Kaplan–Meir analysis indicated that high level expression of miR-572 was associated with poorer overall survival (Zhang et al., 2015a). miR-572 also can induce proliferation, invasion and inhibit apoptosis of nasopharyngeal carcinoma cells by targeting protein phosphatase 2 regulatory subunit Bgamma (Yan et al., 2017). miR-661 was observed to be upregulated in non-small cell lung cancer (NSCLC) tissues as compared to paired adjacent tissues. Furthermore, miR-661 promoted proliferation, migration and metastasis of NSCLC by regulating RB1 and mediating epithelial-mesenchymal transition process in NSCLC (Liu et al., 2017). Yamaguchi et al. (2017) observed miR-4521 was downregulated in chemotherapy-resistant renal cell carcinoma. However, the studies on miR-3648/572/661 and miR-4521 remains rare and their regulatory relationship with our predicted target genes have not been investigated.

However, there are some limitations in this study. First, this is a preliminary study to identify the potential carcinogenic mechanisms of amorphous SiNPs. Additional wet experiments are necessary to confirm the expressions of identified genes and miRNAs (i.e., PCR), their interaction relationships (i.e., dual-luciferase, overexpression or knockout in vitro and in vivo) as well as the influence on the cell proliferation, apoptosis, migration and invasion. Second, our obtained miRNAs from the transcriptome array may be pre-miRNAs. Thus, further miRNA microarray (such as Affymetrix GeneChip miRNA v4) or sequencing analysis is essential to screen more crucial mature miRNAs.

Conclusion

Our findings reveal amorphous SiNPs may exert a carcinogenic effect by targeted regulating of miR-3648/572/661 and miR-4521 followed by influencing their downstream target genes (ATP5H/SOD1 and EIF4G2/PABPC1, respectively). These target genes may be involved in cancer development by promoting oxidative stress, translational initiation, while also inhibiting oxidative phosphorylation and NMD. Accordingly, the four miRNAs and their target genes may be underlying biomarkers for prediction of carcinogenesis when exposed to SiNPs and potentially other targets for cancer treatment.

Supplemental Information

Supplemental Information 1 Raw data used in this article.

1.1: GSM2182800.

Click here for additional data file.

Supplemental Information 2 Raw data used in this article.

1.2: GSM2182801.

Click here for additional data file.

Supplemental Information 3 Raw data used in this article.

1.3: GSM2182802.

Click here for additional data file.

Supplemental Information 4 Raw data used in this article.

1.4: GSM2182803.

Click here for additional data file.

Supplemental Information 5 Raw data used in this article.

1.5: GSM2182804.

Click here for additional data file.

Supplemental Information 6 Raw data used in this article.

1.6: GSM2182805.

Click here for additional data file.

Supplemental Information 7 Processed genes in all samples.

Click here for additional data file.

Supplemental Information 8 All differentially expressed genes.

Click here for additional data file.

Supplemental Information 9 Protein-protein interaction data.

Click here for additional data file.

Additional Information and Declarations

Competing Interests

Author Contributions

Data Availability

The authors declare that they have no competing interests.

Dongli Xie conceived and designed the experiments, performed the experiments, analyzed the data, contributed reagents/materials/analysis tools, prepared figures and/or tables, authored or reviewed drafts of the paper, approved the final draft.

Yang Zhou conceived and designed the experiments, authored or reviewed drafts of the paper, approved the final draft.

Xiaogang Luo conceived and designed the experiments, contributed reagents/materials/analysis tools, prepared figures and/or tables, authored or reviewed drafts of the paper, approved the final draft.

The following information was supplied regarding data availability:

Raw data is available in the Supplemental Materials.

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
