# Peer review of "Amorphous silica nanoparticles induce tumorigenesis via regulating ATP5H/SOD1-related oxidative stress, oxidative phosphorylation and EIF4G2/PABPC1-associated translational initiation"

_PeerJ, doi:10.7717/peerj.6455_

## Round 0.1 · original submission · Major Revisions

All reviewers indicated that additional experiments and/or confirmatory analyses would be needed to validate the observations (Reviewer 1: "should demonstrate whether miR-4734 and miR-4521 directly regulate these genes in this cell lines"; Reviewer 2: "got their conclusion, but without any certification. I think the downstream detection to vertify their conclusion is very necessary."; Reviewer 3: "reviewer suggests some bench experiments. What are the characteristics of normal lung bronchial cells following incubation with silica nanoparticles? Perform qPCR validation of mature miRNA expression levels for those miRNAs of interested in lung bronchial cells +/- nanoparticles."). This is particularly true given the differences to earlier work.

Reviewer 1 ·

Basic reporting

.

Experimental design

Experimental design is well designed .

Validity of the findings

The finding is novel.

Additional comments

Manuscript number: #32867

Title: Amorphous silica nanoparticles induce tumorigenesis via regulating miR-4734 and miR-4521 mediated oxidative stress, innate immune response and translational initiation
Comments to the Author

In this manuscript author investigated the underlying toxic mechanisms of amorphous SiNPs on human lung epithelial cells model by using microarray data. Here, they demonstrated that a total of 323 genes were identified as DEGs from Beas-2B exposed to amorphous SiNPs, including 280 upregulated (containing 12 miRNAs) and 43 downregulated genes (containing 29 miRNAs). Furthermore, author demonstrated that these genes were involved in translational initiation (i.e. RPL24, PABPC1), innate immune response in mucosa (i.e. HIST1H2BK) response to reactive oxygen species (i.e. TXN) and oxidative phosphorylation (i.e. ATP5H) and these five were proven to be hub genes after PPI-module analyses [RPL24 (degree = 16), HIST1H2BK (degree = 15), PABPC1 (degree = 15), TXN (degree = 11) and ATP5H (degree = 11)] and they were overlapped with the target genes of differentially expressed miR-4734 and miR-4521.
With these results, author suggests that amorphous SiNPs may induce tumorigenesis via regulating miR-4734 and miR-4521 mediated oxidative stress, innate immune response and translational initiation.
In general, the manuscript is well written. However, data did not support their conclusion. I have some comments that I believe might help the authors in increasing the impact of this manuscript.

Comments
1. Author should confirm whether amorphous SiNPs regulate above protein expression.
2. Author should demonstrate whether miR-4734 and miR-4521 directly regulate these genes in this cell lines
3. There some typos. Collect all.

Reviewer 2 ·

Basic reporting

no comment

Experimental design

no novelty in the experiment design.

Validity of the findings

I checked the published work by Li et al, 2017, the origin of the raw data. In this paper, the results were somekind opposite to the raw data. Please recheck the data analysis and thus conclusions.

Additional comments

This work was done based on raw data from Gene Expression Omnibus database. The author analyzed the data, and got their conclusion, but without any certification. I think the downstream detection to vertify their conclusion is very necessary. More important to say, the author should recheck their data analysis. For instance, the result of the differently expressed genes was somekind opposite to the published work by Caixia Guo and Yanbo Li, 2017.

Reviewer 3 ·

Basic reporting

The manuscript is clearly written. The numerous abbreviations do make it confusing at times. The reviewer suggests providing a clear definition for each abbreviation used, even if it appears obvious.

More background if needed from the original manuscript and dataset (Guo et al. Nanotoxicology 2017. PMID: 29164963). Specifically, what are the differences/benefits of the authors’ analyses as compared to the published primary analyses? What were the number of differentially expressed genes in both analyses? Why were certain genes excluded? What bioinformatics analyses were performed in both? Compare and contrast the two studies.

The figures are not well labeled. What are the units for the heat map in Figure 1? What is the x-axis in Figure 2B? The interaction networks are not clear. What do they illustrate and how is it relevant?

Experimental design

The authors present new analyses of a published dataset from the Gene Expression Omnibus database (GSE82062). This dataset contains triplicate samples profiled on Affymetrix Human Transcriptome Array 2.0 platform. It appears that the authors used a 4-fold change (|logFC| > 2) instead of a 2-fold change, as originally published, to identify differentially expressed genes. It is not clear what benefits this provides.

It is important to point out that the transcripts labeled as miRNAs in this dataset are not the mature, functional miRNA form but are the transcribed primary (pri-miRNA) or preprocessed hairpin (pre-miRNA). As an example, MIR4734, with transcript ID TC17001428.hg.1 contains 30 probes of varying lengths and sequences that recognize the transcripts from chr17:36858515-36858583 (-). To interrogate miRNA expression, a miRNA-specific platform, such as the Affymetrix GeneChip miRNA v4.

While the authors do provide the packages used for data analysis, the versions are omitted in some cases. In addition, the reviewer suggests including the R scripts used as a supplement.

Validity of the findings

As presented, it is unclear to the reviewer how these analyses compare to and improve upon the primary manuscript (Guo et al. 2017).

Of note, the published primary analyses (Guo et al. 2017) found 821 differentially expressed genes with 5 upregulated and 816 downregulated. However, in the analyses presented here, the authors identified 280 upregulated genes and 43 downregulated genes. Please address this discrepancy.

What is the biological relevance of the protein-protein interaction networks?

As stated in 2. Experimental design, the reviewer is concerned that miRNA-based conclusions are overstated. Since only the pri-miRNA or pre-miRNA are interrogated, no information is available for the mature, functional miRNA. Although the mature miRNA may be differentially expressed following silica nanoparticle treatment, this cannot be garnered from the current expression data.

Additional comments

In addition to clarifying those items mentioned in 1. Basic reporting, 2. Experimental design, and 3. Validity of the findings, the reviewers has the following suggestions:

Show all differentially expressed genes in the heat map of Figure 1. Identify clusters and perform bioinformatics analyses on those genes within each cluster. A table of these expression data, including replicate expression, averaged expression, fold change, and p-value, would benefit as a supplement. The currently presented Table 1 only shows logFC and P-values for the top 25 up or down genes (based on p-value). This is also incorrectly labeled as top 10.

Relatedly, the data in tables 2, 3, and 4 appears to be on a subset of the entire analysis results. What was the rationale in selecting data for each table? The reviewer suggests including the complete DAVID results as a supplement.

To improve the current manuscript, the reviewer suggests some bench experiments. What are the characteristics of normal lung bronchial cells following incubation with silica nanoparticles? Such phenotypic assays are proliferation, apoptosis, migration, and invasion. Perform qPCR validation of mature miRNA expression levels for those miRNAs of interested in lung bronchial cells +/- nanoparticles.

The reviewer is curious about the composition of the the SiNPs used. How do these specific silica nanoparticles compared to the multitude of others being used? Are they PEGylated or have other modifications or cargo? More detail on the particles would be appreciated.

---

## Round 0.2 · accepted · Accept

Thank you for your efforts in addressing reviewer comments and revising your manuscript accordingly. Although I was unable to get the original reviewers to re-review this revised version, my analysis indicates that you have addressed their concerns and responded appropriately.